# GERARD: GEneral RApid Resolution of Digital Mazes Using a Memristor Emulator

**Pablo Dopazo** [1], **Carola de Benito** [1,2], **Oscar Camps** [1], **Stavros G. Stavrinides** [3] **and Rodrigo Picos** [1,2,*]

1 Industrial Engineering and Construction Department, Balearic Islands University, 07122 Palma, Spain; pablo.dopcor@gmail.com (P.D.); carol.debenito@uib.es (C.d.B.); oscar.camps@uib.es (O.C.)
2 Health Institute of the Balearic Islands (IDISBA), 07122 Palma, Spain
3 School of Science and Technology, International Hellenic University, 57001 Thessaloniki, Greece; s.stavrinides@ihu.edu.gr
* Correspondence: rodrigo.picos@uib.es

**Abstract:** Memristive technology is a promising game-changer in computers and electronics. In this paper, a system exploring the optimal paths through a maze, utilizing a memristor-based setup, is developed and concreted on a FPGA (field-programmable gate array) device. As a memristor, a digital emulator has been used. According to the proposed approach, the memristor is used as a delay element, further configuring the test graph as a memristor network. A parallel algorithm is then applied, successfully reducing computing time and increasing the system's efficiency. The proposed system is simple, easy to scale up and capable of implementing different graph configurations. The operation of the algorithm in the MATLAB (matrix laboratory) programming enviroment is checked beforehand and then exported to two different Intel FPGAs: a DE0-Nano board and an Arria 10 GX 220 FPGA. In both cases, reliable results are obtained quickly and conveniently, even for the case of a $300 \times 300$ nodes maze.

**Keywords:** memristor; memristive grid; maze solving; shortest path; programmable devices





## 1. Introduction

Since ancient times, humankind has tried to solve labyrinths or mazes. The paradigm of maze solving is found in the Greek myth of Ariadne, who used a thread to help Theseus get out of Minotaur's labyrinth. Today, maze resolution can have multiple applications, as in robotics, topology and many areas of science and technology [1–3]. Graph theory is used as an element to define the maze problem, where optimized path-solving algorithms could then be applied. Some algorithms simply obtain an exit path, while others optimize it by finding the shortest one. One of the latter is the Dijkstra algorithm [4] that calculates all possible paths to reach a final node beginning from an initial one and then compares the total cost of all of them, eventually keeping with the shortest. This algorithm, as well as all of its alternatives, quantum computing excluded [5], requires a long computation time when dealing with complex graphs. To overcome this, parallel computing becomes a very good alternative in reducing computing time and further improves efficiency [6–8].

One of the trends in high performance computing is the use of arrays of memristors or memristive grid performing parallel computing. Memristors are resistive devices whose resistance depends on their dynamical history [9]. In fact, they can be thought of as variable resistances capable to remember their past; that is, memristors can be used as a memories [10], as well as computation elements. One of the applications they have been used in is modeling the distance between nodes in a maze. In this approach, the shortest path between two points corresponds to the current path with the minimum resistance.

The implementation and design of circuits with memristors requires extensive simulations when the number of devices involved is large such as in memories or bio-inspired circuits [11]. Even though there are SPICE implementations of different models [12–16],

in order to speed up simulations, some researchers use digital, analog or mixed-signal emulators [17–24]. The use of these emulators can improve the simulation time, allowing the physical implementation of memristive circuits, while eliminating some undesired effects such as the cycle to cycle variability appearing in ReRAMs (resistive random access memories) [25,26].

In this paper, a fully digital system (under the acronym GERARD: GEneral RApid Resolution of Digital mazes) is implemented that solves mazes in a digital environment by implementing the topology of those mazes as a grid of nodes in a field-programmable gate array device (FPGA), which allows parallel computing. In this proposal, the interconnections between the nodes of the grid are implemented using memristor emulators that are purely digital, as in [27]. Determining the minimum current path is achieved by mapping the distance between nodes to a memristance, which charges a fixed capacitor with a fixed voltage. This way, the time needed for charging the capacitor up to a given voltage provides an estimation of the value of the memristance, thus the length of the path.

The paper is organized as follows. In Section 2, the general method is described. Section 3 explains the algorithm implementation. Section 4 presents the results. Finally, Section 5 discusses the work.

## 2. General Method

The proposed method for the maze solver is based on representing the maze as a matrix of nodes connected through memristors (Figure 1) to its four nearest neighbors, forming a memristive grid as in the original paper of Pershin et al. [8]. The main idea in that paper was measuring the current through the interconnecting memristors, obtaining in this way the minimum current path. That method was based on the capability of memristors to be programmed to a given resistance value using a (relatively) high voltage, while during its normal operation, it could be considered to hold its programmed resistance value.

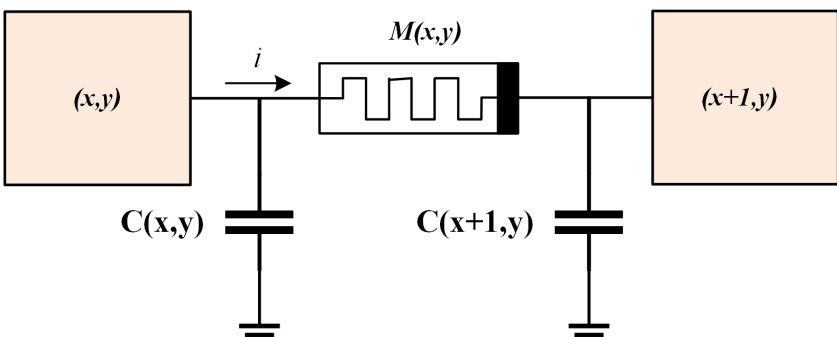

**Figure 1.** Scheme of the general interconnection pattern between two arbitrary adjacent nodes (x,y) and (x + 1) of a N × M grid. *M* and *C* variables denote the programmed resistence and the capacitor, respectively. Notice that, for the sake of clarity, only one of the four connections of each node is shown.

The above-described method had the problem of determining which was the minimum current path, which is a rather complex experimental problem. In this paper, another novel approach is proposed based on measuring the time demanded for a grounded capacitor to be charged through a memristor, as shown in Figure 1. This time obviously depends on the memristance value (in fact, this is a dynamically changing resistance). Since the input is considered to be a constant current, $I_C$, fed through the memristor, its voltage, $V_C$, will be:

$$V_C = \frac{I_C}{C} \cdot t,$$ (1)

where $C$ is the value of the capacitor, and $t$ is the time since the capacitor has started to charge, assuming an initial value for $V_C(t = 0) = 0$. The voltage drop through the

memristor is $V_M = I_C M$, with $M$ being the programmed resistance value of the memristor. The time $t_C$ needed for these two voltages to equate is just:

$$t_C = MC.\tag{2}$$

Thus, for a constant value of $C$, measuring $t_C$ allows for directly estimating the value of $M$. This way, the minimum time is used to calculate the minimum resistance current path, instead of the maximum current. As a side comment, it is worth noticing that a similar effect could be achieved by using a complementary configuration with fixed resistors and memcapacitances.

The flow diagram of the algorithm is shown in Figure 2. The time $t_{(x,y;x+1,y)}$ needed for a signal to propagate from an initial node $(x, y)$ to a destination node $(x + 1, y)$ (Figure 1) is calculated by Equation (2). Once the signal propagates, the destination node is activated and performs a series of actions:

1. It stops listening to any other input, so no other signal can trigger it.
2. It identifies and stores the triggering input port.
3. It propagates the signal to all its non-activated ports.

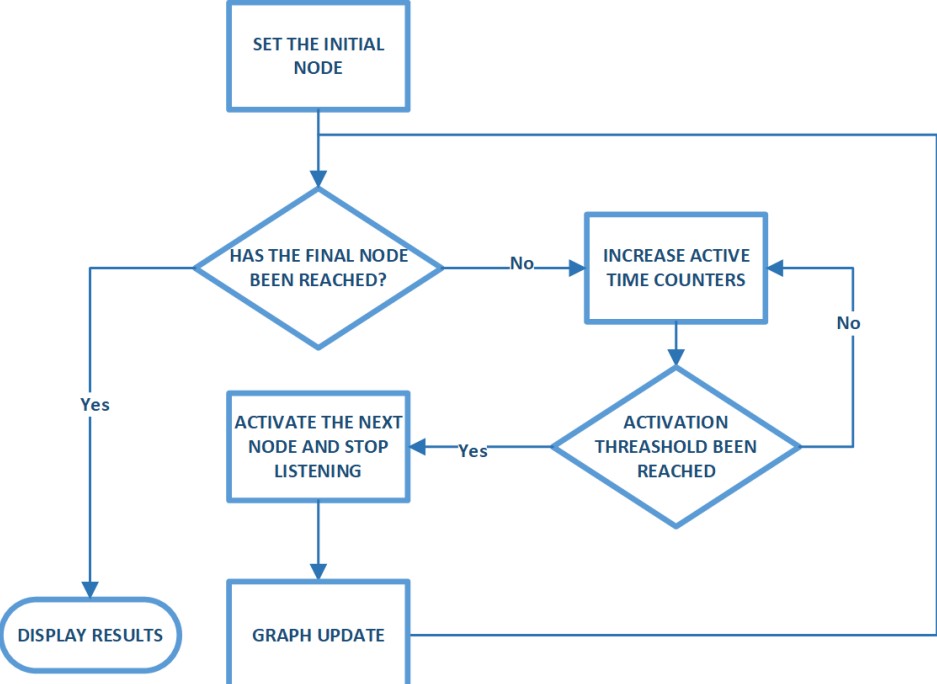

**Figure 2.** The flow diagram of the proposed shortest path algorithm.

Once the final target node is reached, it sends a signal to the controller, which, in turn, recovers the path. This is achieved by recovering the activated input from each node and working the way back from the final node to reconstruct the actual path.

Consequently, the time the algorithm needs to reach the solution is determined by the length of the path and the cost between the nodes in the path, as described in Equation (3):

$$t_{\text{total}} = \sum t_{(x,y;x',y')} = C \sum M_{(x,y;x',y')},\tag{3}$$

where the summation is performed over the nodes $(x,y)$ and $(x',y')$ in the shortest path, and $M_{(x,y;x',y')}$ is the resistance between them.

### 3. Algorithm Implementation

*3.1. Memristor Model Implementation*

There are many models proposed in the literature that reproduce the electrical behavior of memristors. Just for historical reasons, it is worth mentioning that the very first model was proposed in [28], in the same paper that revealed the discovery of actual memristors. It was based on ionic diffusion and received many posterior improvements (see, e.g., [29–32]). Another approach is using charge and flux, as proposed in [33]. Following this approach, some models have also been proposed [34–37], and many more models can be found in the literature; for reviews, see [38–40] .

Following the charge and flux paradigm, a memristor model based on a purely digital emulator [27] was implemented by us. This emulator implements a simple relation between charge $Q$ and flux $\phi$:

$$Q = M(\phi)\phi. \tag{4}$$

Memresistance $M(\phi)$ is also calculated with the simplest relation:

$$M(\phi) = M_0\phi. \tag{5}$$

Full details of the implementation are provided in [27].

Let us note that only a minor modification was needed to fulfill the requirements for this application. This was adding a switch keeping constant the value of the memresistance or allowing it to be programmed, as discussed in the section above. With this modification, once the memristor is programmed, it behaves as an element with a constant resistance.

*3.2. MATLAB Implementation*

The operation of the system developed in the MATLAB programming language implements the flow diagram appearing in Figure 2 and performs the following operations:

1. Program all memristors with a memristance value $M$ corresponding to the distance between nodes.
2. Set the starting point, taking into account that the bottom right element is the end by default (without any loss of generality).
3. Start counting with the first node and propagate the signal to its neighbors with a delay given by Equation (2).
4. When a node receives an input signal, it is marked as active and treated as a new starting point.
5. Repeat from step no. 3 until the final node is reached.
6. If the end node is reached, a signal that the process is finished is sent to the control unit, and the shortest path is then retrieved.

It is noted that the MATLAB implementation of the designed algorithm has been validated against the Dijkstra algorithm [4] up to an $8 \times 8$ matrix, providing exactly the same results. An example is shown in Figure 3 for a simple $3 \times 3$ matrix, where the numbers between the nodes correspond to the distance between them. The green color defines the calculated shortest path (which is the same both by Dijkstra and by the proposed algorithm).

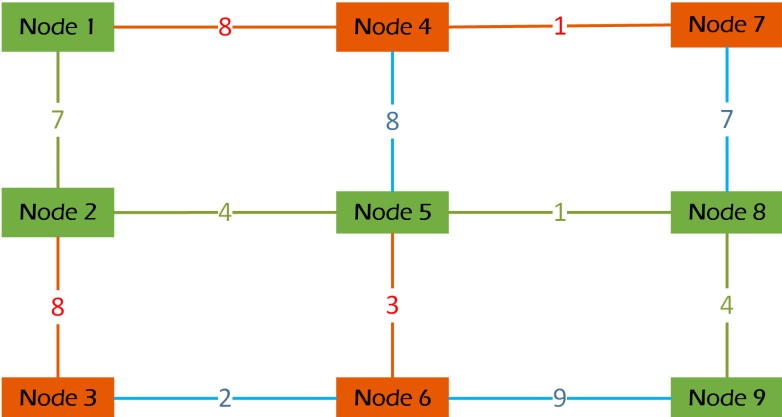

**Figure 3.** The solution in the case of a 3 × 3 example. Notice that the numbers between the nodes represent the resistance. The shortest path is the one passing through the green nodes. The initial node is node 1, and the final node is 9.

### 3.3. Programmable Device Implementation

The FPGA implementation of this novel maze solver consisted of three different parts, namely, the control system, an interconnection element including the memristor emulator and the nodes themselves, as illustrated in Figure 4. Note that the number of cells and memristor-blocks depends on the grid size, since there is an one-to-one correspondence between the physical modules and the maze net.

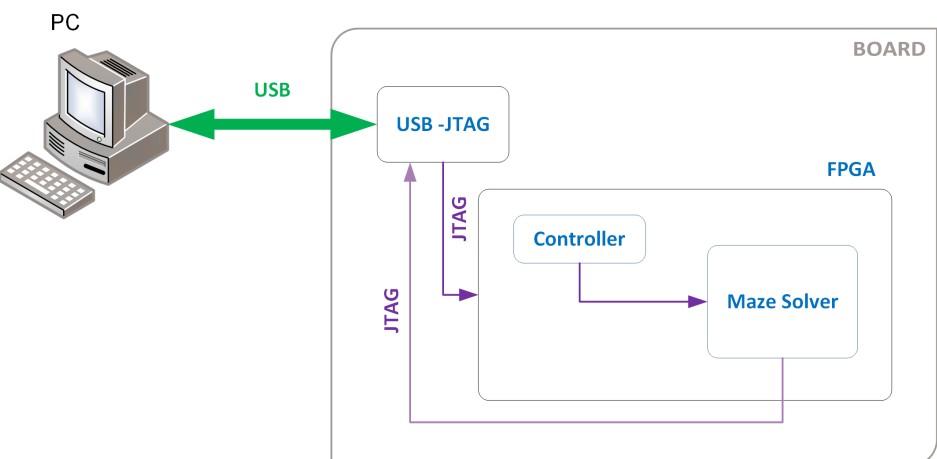

**Figure 4.** Illustration of the system, including the PC running the external software, the USB-to-JTAG interface on the electronic board, and the FPGA, where the general controller and the maze solver are located. See text for details.

### 3.3.1. Communications Block

This block is responsible for the communication between the programmable device and the external systems, in this case, a personal computer (PC). The communications part between the computer (MATLAB) and the FPGA was implemented using a JTAG on-chip instrumentation standard interface, as in [41], where a full description of the procedure is provided. This part consisted of two standard blocks: the vJTAG (virtual JTAG) component and the vJTAG interface, as shown in Figure 5. The vJTAG component is responsible for receiving the information and injecting it into the vJTAG interface. The later saves the data received in a register and then sends its contents to the other components of the system through a dedicated bus. In addition, these components were responsible for sending the result back to the user. The interface between the user and the maze solver in the FPGA was implemented in MATLAB. This uses the internet protocol suite TCP/IP with a dedicated

socket [41] and was responsible for converting user-commands to binary code. It was also receiving back data from the FPGA, displaying them accordingly.

### 3.3.2. Interconnection Block

This block implements a delay equivalent to the cost needed to traverse it. As mentioned above, this delay module (see Figure 6) implements a memristor-capacitor emulator, as in Figure 1. The system can be programmed to a given delay by setting the memristor to an equivalent value provided by Equation (1) and the actual cost of the maze.

Once the memristor is programmed and one of the input ports of the memristor has been activated, the capacitors are charged using a constant voltage input $V_s$ until a threshold value is reached. For a known value of the capacitor and the memristor, this would be equivalent to determining the value of the current used to reach the threshold value.

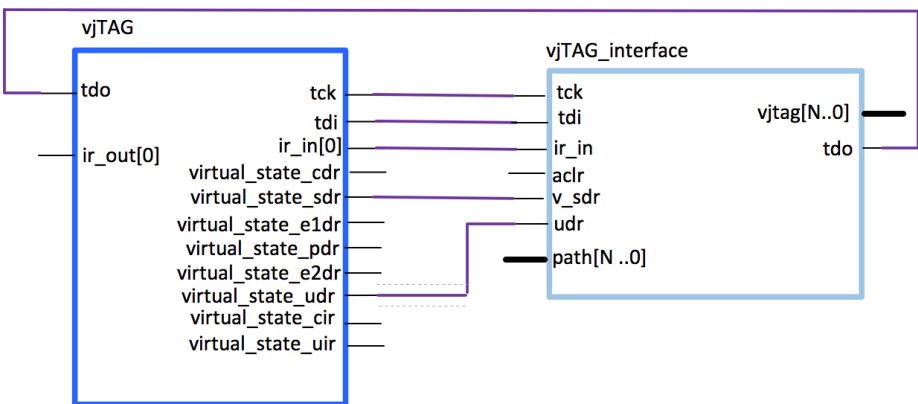

**Figure 5.** The connection between the virtual JTAG (vJTAG) blocks in the communication module, as discussed in [41,42].

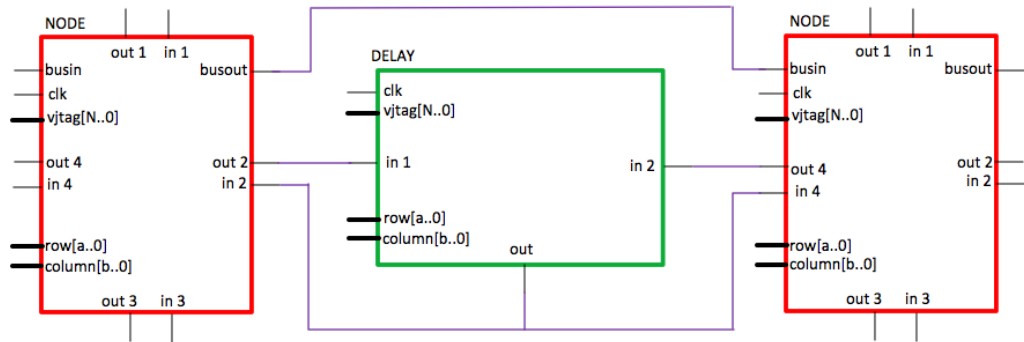

**Figure 6.** The equivalent FPGA implementation of Figure 1, with the nodes and their interconnection. The block labeled "DELAY" is the equivalent of the memristor-capacitor elements, while the "NODE" block corresponds to the (x,y) node elements. All these blocks also show the additional inputs that allow external programming and data recovery, as discussed in the text.

The memristor emulator, implemented in [27], were used as described above, and considered it to be connected to a capacitor at each end, which was initially connected to ground. The capacitor has been implemented as an accumulator with input $i_C$ and output $v_{Co}$. Moreover, the equations have been simplified by setting all the constants of the system to 1, without any loss of generality. At each clock cycle, $v_{Co}(t)$ would be updated as:

$$v_{Co}(t+1) = v_{Co}(t) + i_C C \Delta t. \tag{6}$$

This block will then propagate the signal to the other end by activating the out terminal, when $t = M$, as determined by Equation (2). Notice that the block must only accept the

first input that reaches it (either in1 or in2), and any subsequent input signal has to be disregarded. In this module, the numbered inputs and the output are connected to the nodes, while all other entries follow the same logic as the node. This module uses the clocking (clk) fall edge to create a small delay between the components, allowing the node to update its signals before the delay starts to work. In addition, inside each node and memristor, there was also a control unit connected to the visual joint test action group vJTAG[N..0] bus input, implemented as a state machine, with the corresponding part of the set of instructions shown in Table 1 that allows it to be programmed to the initial value. In order to use a single template component, we have also added two inputs setting the block row and column that identifies the block for programming purposes.

**Table 1.** Set of instructions for the control system. Notice that instructions 010 and 011 need additional arguments (target row and column) to function.

| Code | Description | Parameters |
|------|-------------|------------|
| 001 | Reset all the internal registers | - |
| 010 | Program the value of a memristor | Row, column |
| 011 | Set the starting point | Row, column |
| 100 | Start the process | - |
| 101 | Get the calculated path | - |

### 3.3.3. Node Element

The node components represent each of the network nodes and were connected according to the pattern established previously in MATLAB. These interconnections were made using the intermediate components that generate the signal delay, i.e., the memristor and capacitor emulator described above. Notice that each block has four *in* and four *out* terminals, numbered clockwise from the top, that connect to the delay block as shown in Figure 6.

The numbered inputs and outputs of the node block are used to form an (N × M) graph, corresponding to the actual maze, and are connected to the delay modules. Let us repeat here that the maze path is defined by the interconnections between these nodes. The vJTAG entry is the input for the data sent by the communications module through a shared, read-only bus, whereas the row and column entries mark the position of the component within the system for programming purposes. The modules accepts the corresponding programming instructions shown in Table 1. These instructions allow the user to use the communications block to set the starting point in a specific node, defined by its position, and, once it is set, to start the process of finding the shortest path between this node and the (N,M) node. Finally, the *busin* input and *busout* outputs are connected between each pair of nodes to return the path through the JTAG interface.

## 4. Results

Having in mind the global operation that has been described, this was implemented as follows: When the initial node becomes activated by the user, it activates its four outputs in order to start the counter of all four delays connected to it; then, when the assigned weight value has been reached, the delay activates its output, resulting in turning on the node on the opposite side. An activated node saves in its internal register the one out of the four entries that launched the activation, stopping at the same time to listen to the other entries, which are now transformed into outputs. Then, the node sends a pulse through these new outputs, that propagates in the same fashion until the end node is reached. By this approach, there are several counters running in parallel, achieving the goal of reducing calculation time, thus improving efficiency. As explained above, each node stores the information of the direction from where the first pulse reached it. This information is

sent through a bus connecting each and every node up to the communications module. The later concatenates each of the bits it receives, forming an N-bit vector that is sent to the user via the vJTAG interface (*path[N..0]* input).

Initially, the proof of a concept of the proposed algorithm was checked using the implementation of a 4 × 4 maze example, as shown in Figure 7. In this case, the system needed 16 node and 24 delay modules to implement the desired grid into the DE0-Nano FPGA board, using a 49-bit vJTAG vector and 3-bit row and columns vectors.

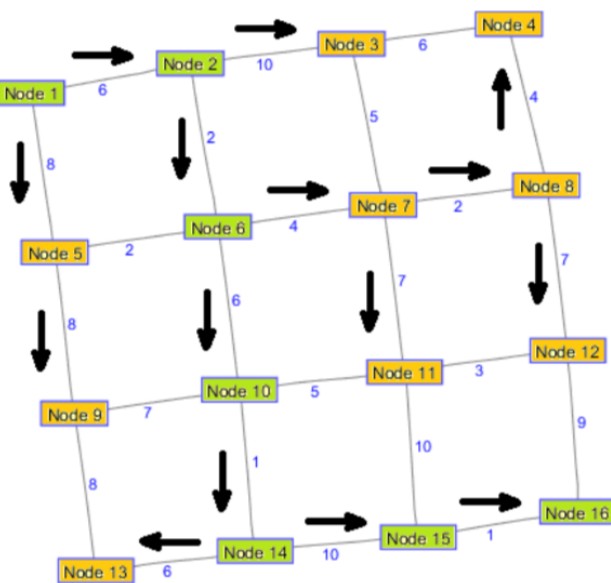

**Figure 7.** The FPGA's returned result in the case of a 4 × 4 maze. The corresponding recovered shortest path (nodes in green) appears, showing the directions in Figure 8, according to Table 2 .

In this example, the memristive grid defining the maze was initially described in MATLAB, and then programmed into GERARD through the dedicated interface. As described above, once the solver was programmed, the signal propagated to the end node and the system returned a signal, which, here was a 49-bit vector containing the direction of the incoming signal for each node according to the Table 2 code. Notice that each node used three bits and these were reorganized in a 4 × 4 array, as shown in Figure 8. Once the result vector was obtained, the user interface decoded it by working its way backwards, from the end node back to the initial w node, obtaining the result shown in Figure 7. In this specific case (a 4 × 4 matrix), the design required a total of 6142 elements (28% of the total), with 6033 combinational functions (27%) and 3274 dedicated logic registers (15%), using a clock frequency of 50 MHz.

| "000" | "100" | "100" | "011" |
| "001" | "001" | "100" | "100" |
| "001" | "001" | "001" | "001" |
| "010" | "001" | "100" | "100" |

**Figure 8.** Array obtained from FPGA for the example graph, indicating the first activating input for each node.

**Table 2.** Incoming signal direction codes.

| Code | Description | Code | Description |
| --- | --- | --- | --- |
| 000 | Initial node | 111 | Final node |
| 001 | Above | 011 | Below |
| 010 | Right | 100 | Left |

Finally, another example was implemented into an Arria 10 GX 220 FPGA card at 200 MHz using the MATLAB FPGA-in-the-loop (FIL) methodology. In this example, the FPGA has been used to speed-up the parallel calculations, and a 300 × 300 maze was generated, using a total of 196,840 logic elements (89%), 68,654 ALM (85%), 301,368 registers (93%), 10,442 Kb of M20K memory (88%) and 1612 Kb (95%) of the MLAB memory. The resistance between the w nodes was generated using 150 2-dimensional Gaussian distributions with random position, dispersion and height, as shown in Figure 9, where the colors represent the cost. In this same Figure, the red line depicts the shortest path, as returned by the algorithm between the start (left, bottom) and the end points (top, right). The time needed for a full MATLAB implementation (with no FPGA) to solve the circuit was around 1900 s, while the FIL version needed only 82 ms to solve the maze, for a total cost of 81,630 (the total resistance, in arbitrary units) for a path length of 608 cells. Retrieving the shortest path thus required 1800 bits. A comparison with the Dijkstra algorithm was not possible using the MATLAB built-in algorithm, since it ran out of memory.

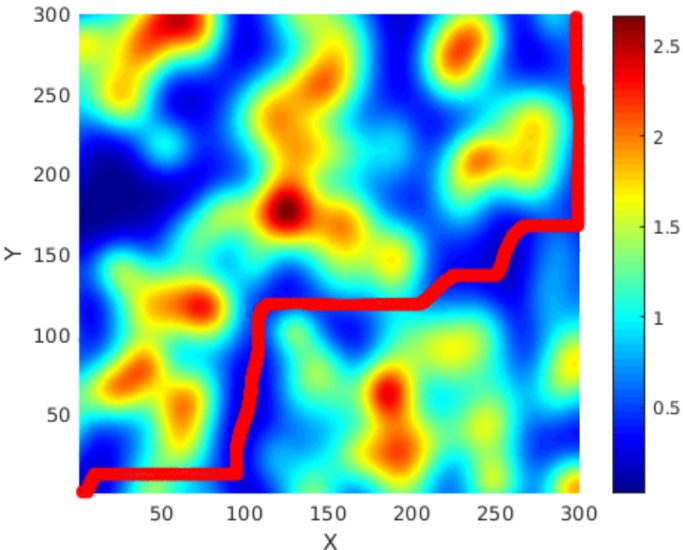

**Figure 9.** An illustration of the results returned in the case of a 300 × 300 maze, used for testing the implementation of the solver, appears. The colors represent the cost, which is the resistance between paths, as indicated in the right bar in arbitrary units, while the red line shows the returned shortest path.

## 5. Conclusions

In this paper, an inherently parallel computation algorithm is demonstrated. It was initially designed in MATLAB, then implemented in a programmed device (FPGA) using a memristor emulator and returned reliable results, equivalent to those obtained using Dijkstra's algorithm. It took the profit of the study of multiple paths in parallel, showing how the fully digital sytem GERARD helps Ariadne to determine the way out of a maze. Two different examples were demonstrated, one with a 4 × 4 matrix and another using a 300 × 300 matrix, both working in a very straightforward way.

The proposed design is simple and easy to scale up for implementing different graph configurations and has been checked with many other examples and using Dijkstra's algorithm [4]. The scalability of the system is limited only by the size of the FPGA. Overcoming this, a proper partitioning scheme could be also utilized. Finally, once actual memristor devices are finally out as a mainstream technology, they could be actually used to implement the proposed maze solver, paving the way for their use in autonomous robotics, among other possible fields.

**Author Contributions:** Conceptualization, C.d.B. and R.P.; formal analysis, C.d.B., O.C., S.G.S. and R.P.; investigation, P.D., C.d.B. and O.C.; methodology, S.G.S.; supervision, R.P.; validation, P.D. and S.G.S.; visualization, P.D. and S.G.S.; writing—original draft, P.D., O.C., S.G.S. and R.P.; writing—review and editing, O.C., S.G.S. and R.P. All authors have read and agreed to the published version of the manuscript.

**Funding:** R.P., C.dB., O.C. wish to acknowledge support from DPI2017-86610-P, TEC2017-84877-R projects, awarded by the MICINN and also with partial support by the FEDER program.

**Data Availability Statement:** Not applicable.

**Conflicts of Interest:** The authors declare no conflict of interest. The funders had no role in the design of the study; in the collection, analyses, or interpretation of data; in the writing of the manuscript, or in the decision to publish the results.

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
