# Peer review of "GERARD: GEneral RApid Resolution of Digital Mazes Using a Memristor Emulator"

_2624-8174, doi:10.3390/physics4010001_

Round 1

Reviewer 1 Report

Why do you not use model of commercial memristor KnowM?

Author Response

Q: Why do you not use model of commercial memristor KnowM?

A: We thank the reviewer for the suggestion. If we had aimed at simulating an actual grid, we could have used it, but it was not the aim of the work. Our aim was to implement a memristive grid that could be used to solve a maze, so we needed to use as simple a model as we could, and thus we opted for the model explained in the paper.

Reviewer 2 Report

  1. Several equations, related to the description of the applied memristor emulator (state-flux, charge-flux, current-voltage relationships)and brief information could be additionally included.
  2. Try to extend the abstract by several sentences.
  3. A brief comparison of the applied memristor emulator with some of the newest memristor models with activation thresholds and modified window functions could be performed, and related information could be observed in some of the newest papers on memristor modeling, for example:
    1. Mladenov, V., Kirilov, S.,„A memristor model with a modified window function and activation thresholds,“ IEEE International Symposium on Circuits and Systems (ISCAS), 2018, pp. 1-5.
    2. Mladenov, V., "A Unified and Open LTSPICE Memristor Model Library" MDPI Electronics, 2021, Vol. 10, no. 13: 1594. https://doi.org/10.3390/electronics10131594.
  4. At the end of the Introduction, a brief description of the following sections could be presented, for example, “Section 2 represents …, In Section 3 is given …”
  5. The keywords “memristive grid” and “programmable devices” could not be found in the text, please include them.

Author Response

We thank the reviewer for the comments. All the improvements appear in red in the manuscript. Below are the specific answers.

Q: Several equations, related to the description of the applied memristor emulator (state-flux, charge-flux, current-voltage relationships)and brief information could be additionally included.

A: We have added an improved version of the equations, but we have tried not to overlap too much with the paper where we describe the actual emulator used.

Q:Try to extend the abstract by several sentences.

A:  We have expanded it, as suggested.

Q: A brief comparison of the applied memristor emulator with some of the newest memristor models with activation thresholds and modified window functions could be performed, and related information could be observed in some of the newest papers on memristor modeling, for example:

    1. Mladenov, V., Kirilov, S.,„A memristor model with a modified window function and activation thresholds,“ IEEE International Symposium on Circuits and Systems (ISCAS), 2018, pp. 1-5.
    2. Mladenov, V., "A Unified and Open LTSPICE Memristor Model Library" MDPI Electronics, 2021, Vol. 10, no. 13: 1594. https://doi.org/10.3390/electronics10131594.

A: We have a paragraph on the existence of different models and emulators, including those proposed by the reviewer, but we have not entered into many details, since the goal of the paper is not discussing the model, which has been published elsewhere, but its application inside a maze solving system.

Q: At the end of the Introduction, a brief description of the following sections could be presented, for example, “Section 2 represents …, In Section 3 is given …”

A:  We have done it, as suggested.

Q: The keywords “memristive grid” and “programmable devices” could not be found in the text, please include them.

A: We have done it, as suggested.

Reviewer 3 Report

The paper entitled "GERARD: GEneral RApid Resolution of Digital mazes using a memristor emulator" proposed a system of searching for optimal paths and concreted on FPGA. Also, a parallel algorithm is applied to reduce computing time and increase the efficiency. The subject/conclusions are rather interesting, so I recommend to accept this paper. However, there are still some points that need to be improved and discussed in the article.

In the following, I summarize these points:

  • In paper title, “using a memristor emulator” can be “using memristor emulator”.
  • In Abstract, “memristive emulator” is “memristor emulator”?
  • In Introduction, some other simple memristor emulators can be reviewed, such as the memristive diode bridges, see e.g., Bursting oscillations and coexisting attractors in a simple memristor-capacitor-based chaotic circuit.
  • As for how to use the memory performance of memristor to realize path measurement, it is suggested to add a clear and detailed description.
  • In the exploration of the shortest path, has the global been calculated several times, and is it possible to reduce it, why?
  • Can you list the efficiency comparison between this method and other algorithms in order to confirm the improvement of this method?
  • The quality of figures 3 and 7 can be further improved.

Author Response

We thank the reviewer for the comments. All the improvements appear in red in the manuscript. Below are the specific answers.

Q: In paper title, “using a memristor emulator” can be “using memristor emulator”.

A: We thank the reviewer for this suggestion, but we feel that, since we are proposing a specific emulator, it's more correct keeping the "a".

Q: In Abstract, “memristive emulator” is “memristor emulator”?

A: It has been corrected.

Q: In Introduction, some other simple memristor emulators can be reviewed, such as the memristive diode bridges, see e.g., Bursting oscillations and coexisting attractors in a simple memristor-capacitor-based chaotic circuit.

A: We have improved the discussion on memristor emulators, including the above suggestion.

Q: As for how to use the memory performance of memristor to realize path measurement, it is suggested to add a clear and detailed description. In the exploration of the shortest path, has the global been calculated several times, and is it possible to reduce it, why?

A: The global time is only calculated once the signal propagating through the grid reaches the end node.Then, by backpropagation, we calculate the time needed to get back to the starting poing. Thus, it is not possible to reduce the number of times it is calculated. This is mentioned in the paper, just above Eq. 3: "Once the final target node is reached, it sends a signal to the controller, which in turn, recovers the path. This is simply achieved by recovering the activated input from each node, and working the way back from the final node to reconstruct the actual path."

Q: Can you list the efficiency comparison between this method and other algorithms in order to confirm the improvement of this method?

A: We have tried to compare the proposed method against the built-in Matlab method, which is assumed to be optimized. However, we could not make it to work for the 300x300 grid. We could make it work for the smaller grids presented before, but since they are trivial examples we don't feel this would provide any significant insight into the performance.

Q: The quality of figures 3 and 7 can be further improved.

A: We have changed the figures, and they are improved.

Round 2

Reviewer 1 Report

The authors have addressed all the reviewer comments, and the manuscript could be accepted as it is.

Reviewer 2 Report

The authors took into account the reviewers recommendations.

Reviewer 3 Report

The authors have revised the manuscript, in my opinion, this revision can be accepted.